# Assessment of drug-susceptible and multidrug-resistant tuberculosis (MDR-TB) in the Central Region of Somalia: A 3-year retrospective study

**Mohamed Abdelrahman Mohamed**[1,2‡]*, **Omer Abdikarin Ali**[3], **Aamir Muse Osman**[4,5,6☯], **Mustapha Goni Abatcha**[7], **Abdirahman Abdirizak Ahmed**[2], **Ali Mohamed Ali**[8], **Abdifatah Ahmed Dirie**[1], **Celso José Bruno de Oliveira**[9,10‡], **Abdinasir Yusuf Osman**[1,11], **Shu-Hua Wang**[10,12☯]*, **Rafael F. C. Vieira**[13,14☯]*

**1** Somali National Institutes of Health, Ministry of Health, Mogadishu, Somalia, **2** Faculty of Veterinary Medicine and Animal Husbandry, Somali National University, Mogadishu, Somalia, **3** Faculty of Health Science, East Africa University, Galkaio, Somalia, **4** Somali One Health Centre, Abrar University, Mogadishu, Somalia, **5** Vector-Borne Diseases Laboratory, Department of Veterinary Medicine, Universidade Federal do Paraná, Curitiba, Paraná, Brazil, **6** College of Veterinary Medicine, Abrar University, Mogadishu, Somalia, **7** Veterinary Service Department, Ministry of Agriculture and Natural Resources, Damaturu, Yobe State, Nigeria, **8** Food and Agriculture Organization of the United Nations, Mogadishu, Somalia, **9** Department of Animal Science, College of Agricultural Sciences, Federal University of Paraíba, Areia, Paraiba, Brazil, **10** Global One Health Initiative, The Ohio State University, Columbus, Ohio, United States of America, **11** The Royal Veterinary College, University of London, Hatfield, United Kingdom, **12** Department of Internal Medicine, College of Medicine, The Ohio State University, Columbus, Ohio, United States of America, **13** Department of Public Health Sciences, University of North Carolina at Charlotte, Charlotte, North Carolina, United States of America, **14** Center for Computational Intelligence to Predict Health and Environmental Risks, University of North Carolina at Charlotte, Charlotte, North Carolina, United States of America

☯ These authors contributed equally to this work.
‡ MAM and CJBO also contributed equally to this work.
* research@nih.gov.so (MAM); shu-hua.wang@osumc.edu (S-HW); rvieira@uncc.edu (RFCV)

## Abstract

### Background

Multidrug-resistant tuberculosis (MDR-TB) remains a public health emergency and a threat globally. Although increasing MDR-TB cases have been recently reported in Somalia, limited information is known. This study aims to determine the prevalence of drug-susceptible and MDR-TB in suspected patients referred to the TB Department in Mudug Hospital, Galkayo, Somalia, and identify potential factors associated with MDR-TB.

### Methods

A 3-year hospital laboratory-based retrospective study was conducted by manually reviewing laboratory records of *Mycobacterium tuberculosis* specimens and GeneXpert MTB/RIF results from January 2019 to December 2021 at the reference mycobacteria laboratory department in Mudug Hospital.

### Results

A total of 714 positive GeneXpert-MTB results were identified: 619 (86.7%) were drug susceptible (no Rifampin resistance [RR] detected) and 95 (13.3%) with RR detected or defined

**Data Availability Statement:** All relevant data are within the paper and Supporting Information files.

**Funding:** The authors received no specific funding for this work.

**Competing interests:** The authors have declared that no competing interests exist.

as MDR-TB. Most of the MDR-TB patients were males (71.6%, 68/95) and between the ages of 15 to 24 (31.6%, 30/95). Most isolates were collected in 2021 (43.2%, 41/95). Multivariate analyses show no significant difference between patients having MDR-TB and/or drug-susceptible TB for all variables.

## Conclusion

This study showed an alarming frequency of MDR-TB cases among *M. tuberculosis*-positive patients at a regional TB reference laboratory in central Somalia.

## Introduction

Multidrug-Resistant tuberculosis (MDR-TB) has become a global public health concern, with an estimated 450,000 cases worldwide and a mortality rate higher than cancer [1]. Globally, MDR-TB is responsible for about one-third of all deaths related to antimicrobial resistance [2]. The World Health Organization (WHO) has classified Somalia as a high multidrug-resistant tuberculosis (MDR-TB) burden country [1, 3, 4]. In 2021, Somalia reported an incidence of 43,000 (27,000–62,000) TB cases with a rate of 250 (158–362) per 100,000 population. The MDR-TB incidence in Somalia was 11,000 (6,500–17,000) with a rate of 12 (0.66–24) per 100,000. The MDR-TB is defined as drug resistance to both rifampicin (RIF) and isoniazid (INH) [1]. WHO also defines individuals with RIF resistance (RR) detected by Gene-Xpert MTB/RIF assay as MDR-TB [1]. In 2021, Somalia reported RIF testing for 62% of the bacteriologically confirmed new TB cases and only 39% of the previously treated cases, with 319 laboratories confirmed for MDR-TB and/or RR-TB [4]. Regionally, knowing the clinical epidemiology and risk factors for MDR-TB when evaluating individuals for TB is essential. Accordingly, this study aimed to determine the prevalence of drug-susceptible and MDR-TB in suspected patients referred to the TB Department in Mudug Hospital, Galkayo, Somalia, and identify potential factors associated with MDR-TB.

## Material and methods

### Study design

A hospital laboratory-based study was conducted at the Mudug Hospital in Galkayo, the central region of Somalia. The Hospital has 250 hospital beds with approximately 4,000 inpatient admissions and 21,000 outpatient visits a year and services a catchment area of 60,000 kilometers. In addition to providing general medical services to the community, the hospital also provides specialized TB services, including treatment, isolation, and laboratory diagnosis. The Hospital is also the primary referral center for refugees, asylum-seekers, and internally displaced people (IDPs) in the region and neighborhood regions.

The Mudug Hospital Mycobacteria Laboratory is the one of the main reference laboratories for *Mycobacterium tuberculosis* in the country and is the only laboratory in the central region of Somalia. We conducted a retrospective study evaluating all laboratory *M. tuberculosis* records from 2019 to 2021. Sputum samples were collected using a spot-morning-spot strategy. The samples were examined using the standard Ziehl-Neelsen (ZN) acid-fast bacilli (AFB) staining technique. The Gene-Xpert MTB/RIF (Cepheid, Sunnyvale, CA) assay was performed on all AFB smear positives specimens. Following WHO guidance, individuals with RR detected by Gene-Xpert MTB/RIF assay were classified as MDR-TB [5], and no RR detected

were classified as drug-susceptible TB. The laboratory does not offer *M. tuberculosis* culture or drug susceptibility testing.

## Data collection and analyses

All patients with a TB diagnosis recorded in the Hospital Laboratory for the treatment of TB from 2019 to 2021 were included. Incomplete demographic data and/or negative *M. tuberculosis* test data were excluded from this study (Fig 1).

Demographic, clinical, and laboratory data, such as age, gender, and GeneXpert results, were extracted from the TB laboratory records and transferred to the Microsoft Excel spreadsheet 2021. Two people cross-checked each entry independently to ensure the quality of entered data. Age was categorized into standard age groups (<15, 15–24, 25–34, 35–44, 45–54, 55–64, ≥ 65), and (Young ≤18 years vs. Old >18 years), year were categorized into pre-covid (2019), and in-covid (2020, and 2021), and regions were also categorized into (Central regions and non-central regions). Cleaned data analyses were performed using SPSS Statistics software (IBM Corp, Armonk, NY, USA, version 26). Descriptive statistics were performed and reported as tables. The chi-square test was used to determine the association between putative independent factors and MDR-TB, and results were considered significant when p ≤ 0.05.

## Ethical approval

Institutional ethical permission was obtained from the Research Committee of East Africa University, Somalia (EAU: 10984). The Department of TB, Mudug General Hospital, informed consent was waived because of the retrospective study design.

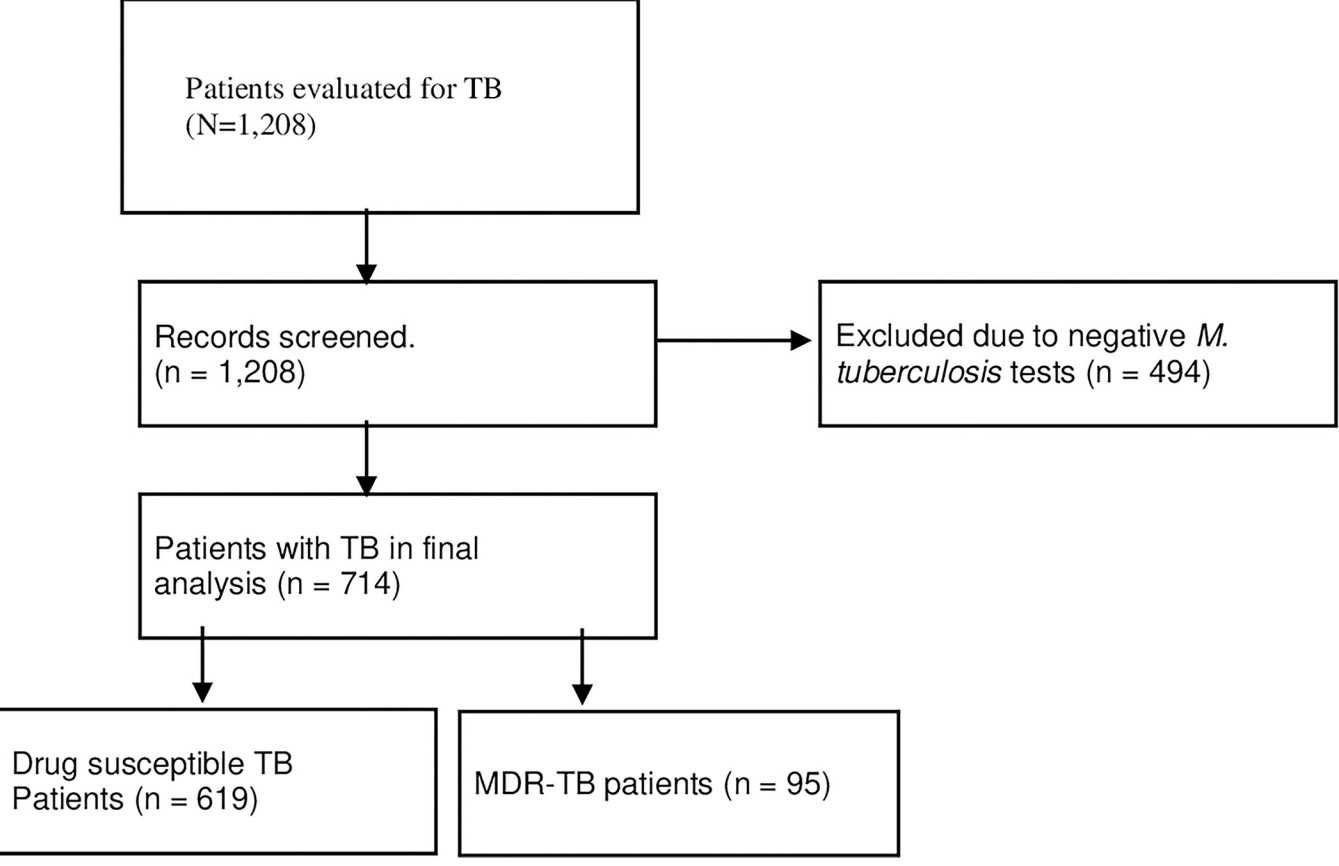

**Fig 1. Flow chart for evaluation of patients with tuberculosis.**

## Results

### Characteristics of the MDR-TB patients

A total of 1,208 samples were collected in the laboratory. Excluding negative *M. tuberculosis* tests and duplicate patients, 714 unique individuals were identified with positive *M. tuberculosis* samples with GeneXpert results. A total of 66.8% were male with a mean age of 34.3, median of 28.0, and range of 1 to 90 years, and 33.2% were female with a mean age of 34.7, median of 30.0, and range of 1 to 80 years of age. Of the 619 drug-susceptible TB (GeneXpert *M. tuberculosis* detected, no RR detected), 72.7% were male, and 28.3% (175/619) were between the ages of 15 to 24 years.

A total of 205 cases in 2020 and 203 cases in 2021 were recorded. GeneXpert RR was detected in 13.3% (95/714; 95% CI: 10.9–16.1%): The majority, 68/95 (71.6%, 95% CI: 61.4–80.4%) were males, 30/95 (31.6%, 95% CI: 22.1–41.9%) were in the age group of 15–24 years, and 41/95 (43.2%; 95% CI: 33.0–53.7%) were collected in the year 2021 (Table 1). All 12 regions reported drug-susceptible TB cases (ranging from 1 to 399). All 12 regions reported drug-susceptible TB cases (ranging from 1 to 399). Four of the 12 regions had 1 to 41 MDR/RR-TB cases. Table 1 shows comparison and distribution of age, gender, years, and region for drug susceptible and drug resistant tuberculosis patients.

Table 2 summarizes the factors associated with drug susceptibility and MDR/RR-TB. Factors such as gender, age (young vs. old), and region (central regions vs. non-central regions) were evaluated, and significant differences were found between having MDR/RR-TB and/or drug susceptibility for age (p = 0.05) and regions (p = 0.002), while no significant differences for gender or year were found.

**Table 1. Comparison between drug susceptible and drug resistant tuberculosis patients.**

| Variable | | Total N = 714 | Drug susceptible- tuberculosis n = 619 (%) | Multi-drug resistant- tuberculosis n = 95 (%) |
|---|---|---|---|---|
| Gender | Male | 518 | 450 (73%) | 68 (68%) |
| | Female | 196 | 169 (27%) | 27 (28%) |
| Age | <15 | 64 | 59 (9.5%) | 5 (5.2%) |
| | 15–24 | 205 | 175 (85.4%) | 30 (14.6%) |
| | 25–34 | 167 | 145 (86.8%) | 22 (13.2%) |
| | 35–44 | 102 | 88 (86.3%) | 14 (137%) |
| | 45–54 | 51 | 44 (86.3%) | 7 (13.7%) |
| | 55–64 | 59 | 50 (84.7%) | 9 (15.3%) |
| | ≥65 | 66 | 58 (87.9%) | 8 (12.1%) |
| Year | 2019 | 241 | 211 (87.6%) | 30 (12.4%) |
| | 2020 | 229 | 205 (89.5%) | 24 (10.5%) |
| | 2021 | 244 | 203 (83.2%) | 41 (16.8%) |
| Regions | Benadir | 18 | 18 (100%) | 0 |
| | Galgadud | 21 | 15 (71.4%) | 6 (28.6%) |
| | Mudug | 440 | 399 (90.7%) | 41 (9.3%) |
| | Bari | 37 | 12 (32.4%) | 25 (67.6%) |
| | Nugal | 61 | 39 (63.9%) | 22 (36.1%) |
| | Somali region | 126 | 125 (99.2%) | 1 (0.8%) |
| | Others | 11 | 11 (100%) | 0 |

Note: Others: Regions with less than 5 admitted patients.

**Table 2. Factors associated with multi-drug resistant (Rifampin resistant) TB and drug susceptibility.**

| Variable | | Drug susceptible n = 619 | | | Drug resistant n = 95 | | |
|---|---|---|---|---|---|---|---|
| | | prevalence (%) | P-value | OR 95%CI: | prevalence (%) | P-value | OR 95%CI: |
| Gender | Male | 450 (73%) | 0.91 (χ2 0.05) | 1.1 (0.7–1.7) | 68 (71.6%) | 0.9 (χ2 0.05) | 0.9 (0.6–1.5) |
| | Female | 169 (27%) | Ref | | 27 (28.4%) | Ref | |
| Age | Old | 518 (83.7%) | 0.05 (χ2 3.97) | 0.5 (0.2–1.0)) | 87 (91.6%) | 0.05 (χ2 3.97) | 2.1 (0.9–4.5) |
| | Young | 101 (16.3%) | Ref | | 8 (8.4%) | Ref | |
| Year | In-covid 19 | 408 (65.9%) | 0.72 (χ2 0.23) | 0.9 (0.6–1.4)) | 65 (68.4%) | 0.72 (χ2 0.23) | 1.1 (0.7–1.8) |
| | Pre-covid 19 | 211 (34.1%) | Ref | | 30 (31.6%) | Ref | |
| Regions | Central regions | 414 (66.9%) | 0.002 (χ2 10.9) | 2.1 (1.3–3.2) | 47 (49.5%) | 0.002 (χ2 10.9) | 0.5 (0.3–0.7) |
| | Non-central regions | 205 (33.1%) | Ref | | 48 (50.5%) | Ref | |

Abbreviation OR: Odds Ratio, CI: Confidence interval, χ2: Chi-square.

## Discussion

Identifying potential risk factors for MDR-TB in a country with a high prevalence of MDR-TB, like Somalia, is essential for TB programs and clinicians. There is limited data from Somalia, and this study provided a three-year review of drug susceptibility and MDR-TB at a reference TB lab in central Somalia. Similar to other studies, more men than women developed MDR-TB in Somalia [5–7]. This is possibly attributed to the fact that men are more likely to consume more tobacco and alcohol, as well as employment outside the home, compared to women, which might also increase their risk of developing TB [8, 9]. Primary MDR-TB is acquired, whereas secondary MDR-TB can develop during treatment. Unfortunately, we do not have a prior history of TB in these patients, which may account for the increased drug resistance.

The higher infection rate recorded in patients between 15–24 years old observed in our study is contrary to the results of other previous studies [7, 10, 11]. This variance-related age could result from geographical and socio-economic factors affecting age. The median age of Somalia is 16 [12]. The potential exposure chances for this age group may be more frequent for study, work, or other social activities. When we dichotomize age into young vs. old, it shows old age is 2.1 times more likely to get MDR than young age.

The COVID-19 pandemic has disrupted routine clinical and public services such as TB care and management. Globally, a decrease in TB cases was reported in 2020 compared to 2021. For Somalia, the WHO reported TB cases for 2019, 2020, and 2021 were 17,000, 17.200, and 17,503, respectively, and for MDR-TB was 299 in 2021 [13]. Our study for central Somalia followed a similar trend, with a decrease in the number of cases in 2020. Alarmingly, the total number of MDR-TB cases was much higher in 2021 compared to 2019 (41 vs. 30).

The overall prevalence of MDR-TB in this study was 9.3% (95/714; 95%). This was comparable to a survey from Somalia reported by Sindani et al. (2013) [8], with a prevalence of 8.9% from 2010–2011. Still, our MDR-TB prevalence was lower than the 51% reported by Guled et al. (2016) [9] from Mogadishu. Other differences besides regional could be attributed to factors such as underlying lung disease, tobacco use, social determinants of health, and socioeconomic characteristics that we did not collect. Regional differences were seen for both drug-susceptible and drug-resistant TB. As expected, more TB cases were reported in the larger cities with a higher population, such as Mudug and Somali regions. Interestingly, more MDR-TB than drug-susceptible TB was reported in the Bari region (25 vs. 12). Further investigation and contact investigation should be conducted to determine the potential for an outbreak.

The laboratory diagnosis of MDR-TB was performed mainly by GeneXpert. The rapid molecular detection of TB and RR-TB has allowed for early treatment and response. However, testing and identifying INH drug resistance is essential when treating drug-susceptible TB. Without knowing INH drug resistance, after the 2-month initiation phase with four drugs (INH, RIF, pyrazinamide, and ethambutol), patients are de-escalated to continuation phase with INH and RIF for four more months. And if INH resistance is present, then the patient is only being treated with RIF, leading to increase resistance and MDR-TB development. Strengthening laboratory capacity is critical in the efforts toward the elimination of TB.

This study's major limitation was the use of secondary data. We could not access the full patient information or prospectively enquire regarding TB risk factors. Important clinical factors include HIV, diabetes, and other co-morbidities, history of prior TB, and contacts with other individuals with TB. Other limitation includes laboratory studies that were limited to GeneXpert results only. We do not know if the patient records were missing additional mono-resistance or poly-resistance. We also do not have the patient follow-up or treatment outcomes. These are all critical information to assist public health TB programs to improve prevention, diagnosis, treatment, and management.

## Conclusion and recommendation

In conclusion, this study has shown an increased incidence of MDR-TB among TB-positive patients at the Regional TB Reference Laboratory in central Somalia. Further evaluations into risk factors and reasons for the regional increase in TB and MDR-TB need to be evaluated to identify any potential outbreaks. Since the Somali community has strong migrant ties, that could have spread MDR-TB more quickly. Early detection of MDR-TB is a critical component of TB elimination programs. Therefore, we recommended that priority be given to strengthening public health surveillance and reference laboratory capacities to improve drug-resistant, MDR, pre-XDR, and XDR-TB diagnosis. Further, we recommend a large prospective multi-center study to evaluate possible clinical factors related to MDR-TB.

## Supporting information

**S1 File. Institutional review board approval document.**
(PDF)

**S1 Data. MDR-TB updated data.**
(XLSX)

## Acknowledgments

We are grateful to the Directors and staff of the Mudug Regional Hospital for granting access to the data. Special thanks also go to the data collectors for their help. AYO is a member of, and acknowledges support from, the Pan-African Network for Rapid Research, Response, Relief and Preparedness for Infectious Disease Epidemics, funded by the European and Developing Countries Clinical Trials Partnership, under the EU Horizon 2020 Framework Programme for Research and Innovation.

## Author Contributions

**Conceptualization:** Mohamed Abdelrahman Mohamed, Omer Abdikarin Ali, Aamir Muse Osman, Shu-Hua Wang, Rafael F. C. Vieira.

**Data curation:** Mohamed Abdelrahman Mohamed, Aamir Muse Osman, Abdirahman Abdirizak Ahmed, Shu-Hua Wang, Rafael F. C. Vieira.

**Formal analysis:** Mohamed Abdelrahman Mohamed.

**Methodology:** Omer Abdikarin Ali, Aamir Muse Osman, Celso José Bruno de Oliveira, Shu-Hua Wang, Rafael F. C. Vieira.

**Visualization:** Aamir Muse Osman.

**Writing – original draft:** Aamir Muse Osman, Abdirahman Abdirizak Ahmed, Shu-Hua Wang, Rafael F. C. Vieira.

**Writing – review & editing:** Mohamed Abdelrahman Mohamed, Aamir Muse Osman, Mustapha Goni Abatcha, Ali Mohamed Ali, Abdifatah Ahmed Dirie, Celso José Bruno de Oliveira, Abdinasir Yusuf Osman, Shu-Hua Wang, Rafael F. C. Vieira.

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
