## [Decision Letter · Decision Letter 0]

11 May 2023

PGPH-D-23-00423

Assessment of drug-susceptible and multidrug-resistant tuberculosis (MDR-TB) in the Central Region of Somalia: A 3-year Retrospective Study

Dear Dr. Wang,

Thank you for submitting your manuscript to PLOS Global Public Health. After careful consideration, we feel that it has merit but does not fully meet PLOS Global Public Health’s publication criteria as it currently stands. Therefore, we invite you to submit a revised version of the manuscript that addresses the points raised during the review process.

We look forward to receiving your revised manuscript.

Kind regards,

Ben Pascoe

Academic Editor

Journal Requirements:

1. Please send a completed 'Competing Interests' statement, including any COIs declared by your co-authors. If you have no competing interests to declare, please state "The authors have declared that no competing interests exist". Otherwise please declare all competing interests beginning with the statement "I have read the journal's policy and the authors of this manuscript have the following competing interests:"

2. Please provide separate figure files in .tif or .eps format only and remove any figures embedded in your manuscript file. Please also ensure all files are under our size limit of 10MB.

3. We do not publish any copyright or trademark symbols that usually accompany proprietary names, eg  ©, ®, ™  (e.g. next to drug or reagent names). Please remove all instances of trademark/copyright symbols throughout the text, including ® on page 9.

4. We have noticed that you have uploaded Supporting Information files, but you have not included a list of legends. Please add a full list of legends for your Supporting Information files after the references list. 

Additional Editor Comments (if provided):

Your manuscript has now been assessed by two independent reviewers and I would like you to revise the manuscript in line with their feedback below.

Both reviewers found the topic of interest and given that very little work on Tb has been published from this region, this study will provide a valuable insight into the burden of multidrug resistant Tuberculosis in Somalia.

However, both reviewers feel that there is significant work to be done before it can be accepted for publication. I would like you to pay close attention to the reviewers comments and revise the manuscript accordingly.

Specifically an updated version of the manuscript should address all concerns regarding the statistical methods used.

Reviewers' comments:

Reviewer's Responses to Questions

**Comments to the Author**

1. Does this manuscript meet PLOS Global Public Health’s publication criteria? Is the manuscript technically sound, and do the data support the conclusions? The manuscript must describe methodologically and ethically rigorous research with conclusions that are appropriately drawn based on the data presented.

Reviewer #1: Partly

Reviewer #2: No

2. Has the statistical analysis been performed appropriately and rigorously?

Reviewer #1: No

Reviewer #2: No

3. Have the authors made all data underlying the findings in their manuscript fully available (please refer to the Data Availability Statement at the start of the manuscript PDF file)?

Reviewer #1: No

Reviewer #2: No

4. Is the manuscript presented in an intelligible fashion and written in standard English?

Reviewer #1: Yes

Reviewer #2: Yes

5. Review Comments to the Author

Reviewer #1: The author tried to evaluate the prevalence of MDR-TB in the Central Region of Somalia based on a single-center 3-year laboratory records of GeneXpert MTB/RIF results. A total of 714 unique individuals was finally enrolled and among these, 95 were defined as MDR-TB. In general, this research article was rather small with quite featureless results. The collected baseline data just included gender, age and region. More factors associated with MDR-TB, such as history of prior TB, close contact with TB, other lung disease, co-morbidities (HIV infection, diabetes), tobacco or alcohol use and so on, should be taken into account. In addition, the laboratory results were limited to GeneXpert results only. Drug susceptibility testing to indicate drug resistance other than rifampin should also be carried out.

Reviewer #2: This is an interesting study but falls short in analysis. I suggest the following major revisions be done:

1. Plot trends in DS-TB separately froo trends in MDR-TB from 2019 to 2012 and stratify them according to all the predictor variables: gender, age, year, and regions ( these are 8 plots, you could put them in a panel)

2. Perform chi-squared tests as you did, but in each of the two classes (separate for DS-TB and separate for MDR-TB) - these will be two tables. Try using Fisher’s test in case the chi-square test fails

3. Issues with logistic regression: First remember you sampled hospital records and so the sample size divisions between those with DS-TB and MDR-TB were not pre-determined in a sample size collection. The sample size is unbalanced, and this must be the reason why you get non-significant results. This is because logistic regression fits a Maximum Likelihood Estimate by minimising an objective function which is evaluated at all the data points. If the data is unbalanced, then the minimisation will be unbalanced too. In your car, there are 95 1s (for MDR-TB+ ) and 619 0s (for DS or MDR-TB-). Another way is to take the 95 who are MDR-TB and find their ‘match’, for example, match by gender, age-groups, year and regions as much as possible until you get a more balanced sample ( you could take for each MDR-TB + you take two DS-TB members) and repeat the analysis regression analyses

4. Or instead of logistic regressions, peform analysis of odds ratios/relative risks. You do this by creating a 2 by 2 table in which the outcome (DS-BT vs MDR-TB) is compared to a dichotomized explanatory variable (you could dichotomize age into young vs old; gender is already dichotomized, year could be pre-covid 19 vs in-covid 19, region could be ‘central Somalia vs non-central Somalia)

5. The number and type of predictor variables collected could just by the nature of problem not be the appropriate ones, and that is why you have negative results

6. These suggested analyses will enrich your analysis and probably confirm whether your negative results are really because of an underlying population process and not due to shortcomings in statistical analysis.

6. PLOS authors have the option to publish the peer review history of their article (what does this mean?). If published, this will include your full peer review and any attached files.

**Do you want your identity to be public for this peer review?** For information about this choice, including consent withdrawal, please see our Privacy Policy.

Reviewer #1: No

Reviewer #2: **Yes: **Emmanuel Abraham Mpolya

---

## [Editor Report · Decision Letter 1]

3 Aug 2023

Assessment of drug-susceptible and multidrug-resistant tuberculosis (MDR-TB) in the Central Region of Somalia: A 3-year Retrospective Study

PGPH-D-23-00423R1

Dear Dr. Wang,

We are pleased to inform you that your manuscript 'Assessment of drug-susceptible and multidrug-resistant tuberculosis (MDR-TB) in the Central Region of Somalia: A 3-year Retrospective Study' has been provisionally accepted for publication in PLOS Global Public Health.

Best regards,

Ben Pascoe

Academic Editor

Thank you for incorporating suggestions from the reviewers, especially additional statistical analyses to further interrogate your results.